# The *Epipactis helleborine* Group (Orchidaceae): An Overview of Recent Taxonomic Changes, with an Updated List of Currently Accepted Taxa

**DOI:** 10.3390/plants10091839

**Published:** 2021-09-04

**Authors:** Zbigniew Łobas, Anatoliy Khapugin, Elżbieta Żołubak, Anna Jakubska-Busse

**Affiliations:** 1Department of Botany, Faculty of Biological Sciences, University of Wroclaw, 50-328 Wroclaw, Poland; elzbieta.zolubak@uwr.edu.pl; 2Institute of Environmental and Agricultural Biology (X-BIO), Tyumen State University, 625003 Tyumen, Russia; hapugin88@yandex.ru; 3Joint Directorate of the Mordovia State Nature Reserve and National Park “Smolny”, 430005 Saransk, Russia

**Keywords:** *Epipactis*, Helleborines, morphological species complex, orchids, taxonomy

## Abstract

The *Epipactis helleborine* (L.) Crantz group is one of the most taxonomically challenging species complexes within the genus *Epipactis*. Because of the exceptionally high levels of morphological variability and the ability to readily cross with other species, ninety different taxa at various taxonomic ranks have already been described within its nominative subspecies, but the taxonomic status of most of them is uncertain, widely disputed, and sometimes even irrelevant. The present review is based on results of the most recent research devoted to the *E*. *helleborine* group taxonomy. In addition, we analysed data about taxa belonging to this group presented in some research articles and monographs devoted directly to the genus *Epipactis* or to orchids in certain area(s). Based on the reviewed literature and data collected in four taxonomic databases available online, we propose an updated list of the 10 currently accepted taxa in the *E*. *helleborine* group (two species, six subspecies, and two varieties), which includes *E*. *helleborine* (L.) Crantz subsp. *helleborine*; *E*. *helleborine* subsp. *bithynica* (Robatsch) Kreutz; *E*. *helleborine* subsp. *distans* (Arv.-Touv.) R.Engel and P.Quentin; *E*. *helleborine* subsp. *neerlandica* (Verm.) Buttler; *E*. *helleborine* var. *tangutica* (Schltr.) S.C.Chen and G.H.Zhu; *E*. *helleborine* subsp. *tremolsii* (Pau) E.Klein; *E*. *helleborine* subsp. *voethii* (Robatsch) Jakubska-Busse, Żołubak, and Łobas, stat. nov.; *E*. *condensata* Boiss. ex D.P.Young; *E*. *condensata* var. *kuenkeleana* (Akhalk., H.Baumann, R.Lorenz, and Mosul.) Popovich; and *E*. *cupaniana* C.Brullo, D’Emerico, and Pulv.

## 1. Introduction

The long and turbulent history of changes in taxonomy of the genus *Epipactis* Zinn, 1757 (Orchidaceae) is well documented [1,2,3,4,5,6,7,8,9,10,11,12]. The most widely contested aspect of its taxonomy is whether to treat many of its morphologically different but poorly defined taxa as new species or if is more appropriate to transfer some of them to lower taxonomic ranks, such as subspecies, variety, or form [13,14]. The currently adopted classification system of *Epipactis* does not take sufficient account of the variation range of morphological characters within its taxa, nor does it explain the underlying sources of this variability. Moreover, the species delimitation within *Epipactis* is often complicated by the ease with which the interspecific hybrids are formed in the locations inside the native ranges of the parental species, the existence of which is a common cause of taxa misidentification [15,16,17,18,19,20,21]. The fluidity of the morphological boundaries between various *Epipactis* species causes serious difficulties in determinig the diagnostic characters useful in species identification [22,23,24]. As a result, there is still no official account of the species included within the genus *Epipactis*. The estimates range from a few to several dozen depending on the source (e.g., [6,17,25,26,27,28,29,30,31,32,33,34,35,36,37,38]).

The primary aim of taxonomic research is to provide a comprehensive classification system, which reflects the observed relationships between the taxa at the morphological, geographical, and genetic levels [39]. The major impediment in achieving a taxonomic consensus within the genus *Epipactis* is its exceptionally high morphological variability, which is still insufficiently documented and requires further clarification [11,14]. The main source of this variability, referred to the phenotypic plasticity, is manifested in natural populations of many *Epipactis* species through the slight differences observed in the vegetative (e.g., shoots and leaves) and generative (i.e., flowers and their separate elements) parts of the individual plants [4,15,40,41,42]. This broad spectrum of morphological variation also provided a wide range of characters that delimitate and group the species within the genus *Epipactis* over the past few decades (e.g., [14,17,21,23,28,29,30,31,32,43,44,45,46,47,48,49,50,51,52,53,54]). As a consequence, a large number of morphologically similar species and infraspecific taxa (i.e., subspecies, varieties, or forms), usually of local or narrowly restricted occurrence, have been described within *Epipactis* [17,23,31,55]. However, the taxonomy and systematic position of the majority of these taxa are still not well understood and highly debated [4,11,13,24,42,56,57,58,59].

One of these taxonomically problematic species is *Epipactis helleborine* (L.) Crantz, native to Eurasia and North Africa and widely naturalised in North America [27,31,32,58,60]. It is a common cross-pollinating (allogamous) orchid species with a very wide ecological amplitude. It grows in areas with nutrient rich soils and a broad pH spectrum (usually in alkaline conditions) and, unlike the other species of *Epipactis*, has a highly variable habitat preference. Its natural populations are usually found in forests, amongst shrubs, or in partly disturbed vegetation sites, ranging from lowland floodplains to mountain spruce forests up to the altitude of 2200 m a.s.l. [31,37,51,52,53,61,62,63]. It is also increasingly observed in the areas strongly impacted by human activity, such as roadsides, cemeteries, railway embankments, gravel pits, gardens, and urban parks [64,65].

The recently increased interest in the evolutionary history of the genus *Epipactis* has resulted in some significant changes in its taxonomy [11]. The most important and widely challenged one of them is considered the present division of this genus into nine morphologically distinct species complexes [66,67,68,69]. Among them, there is also a group devoted to *E. helleborine*, for which the circumscription has already been reorganized by numerous scientists [3,5,8,10,17,23,30,31,47,70,71]. Interestingly, the taxonomic status of individual taxa included in this group is still chaotic and in need of clarification.

Because of the general confusion concerning the taxonomy of the genus *Epipactis*, caused mainly by the frequent changes in its infrageneric classification, we aimed to present here an updated list of the 10 currently accepted taxa included in the *E*. *helleborine* group. As a decisive criterion for the selection of individual taxa to our circumscription, we used the results of recent genetic and morphometric analysis in relation to the total 41 taxa that have been included in this group.

## 2. Recent Taxonomic Publications Devoted to the *Epipactis helleborine* Group

So far, a number of the research articles and monographs have been published by representing the description and taxonomic treatments of taxa of the *Epipactis helleborine* group [3,5,8,10,17,23,30,31,47,70,71] (see Table 1). At the beginning, Tyteca and Dufrêne [47] conducted the medium-scale biostatistical study of the genus *Epipactis* focused on only seven allogamous species (autogamous taxa were explicitly excluded) from the south-western limit of its distribution range in Europe. But the authors concluded that at least five species (i.e., *E*. *helleborine* s.str., *E*. *distans* Arv.-Touv., *E*. *neerlandica* (Verm.) Devillers-Tersch. and Devillers, *E*. *tremolsii* Pau, and *E*. *lusitanica* D.Tyteca) should be included within the *E*. *helleborine* group. Tyteca and Dufrêne [47] also used the results of multivariate analysis of 28 carefully chosen characters of floral and vegetative morphology (particularly the differences in the flower structure) to prove that the four taxa included in this group are sufficiently different from *E*. *helleborine* s.str. and should be treated as independent species rather than at the subspecific rank.

Later, the circumscription of the *Epipactis helleborine* group was delimited by a yet another set of clearly defined morphological characters, including the appearance of the shoot, labellum, ovary, and pedicel. However, the morphometric analysis of these characters was not as detailed as that of the other published taxonomic treatments (e.g., [23,47]). Delforge [30] divided the 23 species belonging to the *E*. *helleborine* group into three subgroups, i.e., the *E*. *leptochila* subgroup (five taxa), the *E*. *helleborine* subgroup (13 taxa), and the *E*. *tremolsii* subgroup (five taxa). Within the *E*. *helleborine* subgroup, this author included *E*. *helleborine* s.str. and 12 other morphologically similar species (Table 1). This subgroup was also further sub-divided into two additional sections: one with the cross-pollinating species and another with autogamous taxa only.

The increase in number of new taxa described within the genus *Epipactis* has led to some significant changes in its infrageneric classification. As a consequence, two new characters were added by Delforge [17] to the circumscription of the *E*. *helleborine* group, i.e., the leaf and the inflorescence morphology. Four of the previously used characters, i.e., the appearance of the shoot, labellum, ovary, and pedicel have also been redefined. Thus, Delforge’s newly circumscribed *E*. *helleborine* group included 13 taxa (11 at the rank of species and two varieties). Six of which were included in the author’s previous study [30], where one (i.e., *E*. *helleborine* var. *youngiana* A.J.Richards and A.F.Porter) Kreutz) has changed its taxonomic rank (Table 1).

Subsequently, Brullo et al. [23] have expanded the *Epipactis helleborine* group by including *E*. *cupaniana* C.Brullo, D’Emerico, and Pulv., a newly described endemic from the mesophilous Holm oak woods in north-central Sicily. Their circumscription of the *E*. *helleborine* group included 11 additional species and was broadly based on the system proposed by Delforge in 2006 (Table 1). The authors also conducted a morphometric analysis of a broad range of 37 characters of floral and vegetative morphology. The obtained results suggest that *E*. *cupaniana* does indeed belong to the *E*. *helleborine* group. This taxon is morphologically and karyologically different from *E*. *helleborine* s.str. and can be accepted as a separate species. Despite this conclusion, Delforge [31] did not include *E*. *cupaniana* in his latest concept of the *E*. *helleborine* group.

One year later, in 2014, *Epipactis condensata* subsp. *kuenkeleana* (Akhalk., H.Baumann, R.Lorenz, and Mosul.) Kreutz, Fateryga, and Efimov was published as a new combination for the species formerly known as *E*. *viridiflora* subsp. *kuenkeleana* Akhalk., H.Baumann, R.Lorenz, and Mosul. [8], where then Delforge raised this latter taxon to full species status (i.e., *E*. *kuenkeleana* (Akhalk., H.Baumann, R.Lorenz and Mosul.) P.Delforge) [72]. Thereafter, *E*. *condensata* subsp. *kuenkeleana* was put into synonymy with the nominative subspecies [10,70]. However, finally, in 2020, plants within the same taxon corresponding to the former subsp. *kuenkeleana* were considered as a phenotypic form, confined to shady forest communities, and described as *E*. *condensata* var. *kuenkeleana* (Akhalk., H.Baumann, R.Lorenz, and Mosul.) Popovich [70].

The most recent taxonomic treatment of the *Epipactis helleborine* group in Europe, North Africa, and the Middle East [31], is broadly based on an earlier account by the same author [17] and expands to comprise 17 taxa, five of which are included here for the first time, and one taxon (i.e., *E*. *pontica* Taubenheim) which is transferred to *E*. *leptochila* group (Table 1).

## 3. List of Names of Infraspecific Taxa in *Epipactis helleborine* and its Current Taxonomic Status

The seemingly endless morphological variation observed across the entire distribution range of *Epipactis helleborine* s.str. is clearly reflected by the list of its infraspecific taxa presented below in Table 2.

As it turns out, in the light of the data collected in four taxonomic databases available online [73,74,75,76], as many as ninety morphologically similar taxa have been distinguished within *Epipactis helleborine* s.str. at various taxonomic ranks since its original description as *Serapias helleborine* L. by Carl Linnaeus in 1753 [33,77,78,79,80,81,82]. Among these, at the ranks of variety and subspecies have been classified respectively 37 and 36 names of taxa, and at the rank of form, 17 have been classified (Figure 1).

It should be noted that most of them (64) were synonymised with *Epipactis helleborine* s.l. almost as soon as they were published and currently only five infraspecific taxa are accepted, i.e., *E*. *helleborine* (L.) Crantz subsp. *helleborine*, *E. helleborine* subsp. *bithynica* (Robatsch) Kreutz, *E. helleborine* subsp. *neerlandica* (Verm.) Buttler, *E. helleborine* var. *tangutica* (Schltr.) S.C.Chen and G.H.Zhu, and *E. helleborine* subsp. *tremolsii* (Pau) E.Klein [3,63,71,83,84]. The remaining 26 published names of taxa do not currently have a taxonomic relationship with *E. helleborine* s.l. (Figure 2).

## 4. Conclusions

Since the genus *Epipactis* has been divided into several species complexes based on morphological characters, more than forty taxa have been classified into the *E*. *helleborine* group (see Table 1). These contain such taxa as *E*. *danubialis* Robatsch and Rydlo, *E*. *greuteri* H.Baumann & Künkele, *E*. *halacsyi* Robatsch, *E*. *leptochila* (Godfery) Godfery, *E*. *muelleri* Godfery, *E*. *naousaensis* Robatsch, *E*. *olympica* Robatsch, *E*. *pontica* or *E*. *purpurata* Sm., which, because of their distinct morphological phenotype, were excluded from it over time and (in some cases) provided a basis for effective distinguishing of other groups. Despite the fact that 15 of these taxa were originally included as separate species, they are being considered currently as three out of the five infraspecific taxa published in *E. helleborine* (i.e., *E*. *helleborine* subsp. *helleborine*, *E*. *helleborine* subsp. *neerlandica*, and *E*. *helleborine* subsp. *tremolsii*). Furthermore, two other infraspecific taxa have been genetically confirmed as well-founded, i.e., *E. helleborine* subsp. *distans* (Arv.-Touv.) R.Engel and P.Quentin and *E. helleborine* subsp. *voethii*, although the latter one still has not been officially distinguished at this rank. As it appears, *E. bugacensis* Robatsch and *E. rhodanensis* Gévaudan and Robatsch have in fact a similarly close genetic relationship with *E. dunensis* (T.Stephenson and T.A.Stephenson) Godfery (originally included in the *E*. *helleborine* group), which, in our opinion, due to its floral morphologies (typical of autogamous taxa) should not be classified in this group. Although *E*. *nordeniorum* Robatsch was for a long time assigned to the *E*. *helleborine* group, as a result of recent genetic analysis it turned out to be molecularly similar to *E. albensis* Nováková and Rydlo, classified in a separate group. Some taxa, such as *E. condensata* Boiss. ex D.P.Young and *E*. *cupaniana,* based on results of a detailed morphological analysis of their floral and vegetative characters, should be retained in the *E*. *helleborine* group, although these taxa are still not included there in the most recently published accounts of the genus *Epipactis*.

The boundaries between individual species within the *Epipactis helleborine* group are unclear, making it difficult to determine reliable taxonomic characters useful in the construction of an identification key which would be unambiguously interpreted by different users. In the light of the scientific literature published worldwide, especially because of the impact of the genetic research on our current understanding of the boundaries between various species of *Epipactis*, we think it is appropriate to maintain the *E*. *helleborine* group, but we propose to update its circumscription to better reflect the taxonomic changes summarised in Table 1 that have occurred for its individual members over the past few decades.

Our proposed taxonomic circumscription of the *Epipactis helleborine* group therefore consists of the following only allogamous taxa: *E*. *helleborine* subsp. *helleborine*, *E*. *helleborine* subsp. *bithynica*, *E*. *helleborine* subsp. *distans*, *E*. *helleborine* subsp. *neerlandica*, *E*. *helleborine* var. *tangutica*, *E*. *helleborine* subsp. *tremolsii*, *E*. *helleborine* subsp. *voethii* (Robatsch) Jakubska-Busse, Żołubak, and Łobas, stat. nov., *E*. *condensata*, *E*. *condensata* var. *kuenkeleana* and *E*. *cupaniana*.

Although the proposed list of taxa in the *Epipactis helleborine* group seems to be appropriate at the moment, we treat it as legitimate until new methods of genetic and morphometric analysis are developed, which would allow more precise definition of the *Epipactis* separate species concept in the future.

### Proposal of a New Status for Epipactis helleborine subsp. voethii

*Epipactis helleborine* subsp. *voethii* (Robatsch) Jakubska-Busse, Żołubak, and Łobas, stat. nov.

Basionym: *Epipactis voethii* Robatsch, Mitteilungen der Abteilung für Botanik am Landesmuseum Joanneum in Graz 21/22: 22 (1993).

Comments: This subspecies differs from typical *Epipactis helleborine* s.str. through few developed clinandrium, as well as the slight differences observed in the morphological characters, i.e., the green colour of stems, leaves, and flowers of the individual plants, which are almost lacking in any violet coloration. In the fruiting stage, taxa can be distinguished by the shape of the seeds: in *E*. *helleborine* s.str. the seeds are worm-like, and club-shaped in *E*. *helleborine* subsp. *voethii*.

## Figures and Tables

**Figure 1 plants-10-01839-f001:**
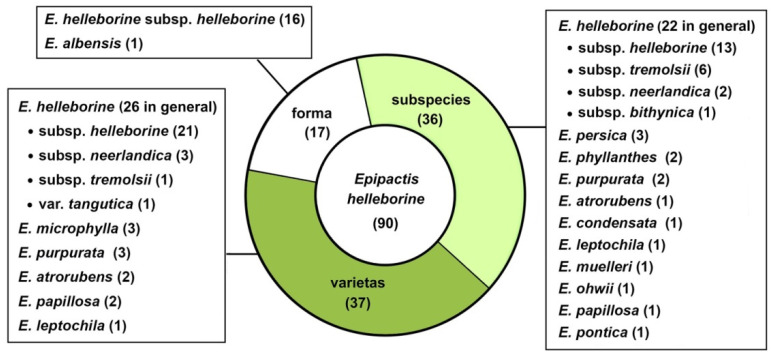
The current status of names of infraspecific taxa published in *Epipactis helleborine* categorised according to their taxonomic ranks, based on [73,74,75,76]. The number of names synonymised with *E. helleborine* s.l. and with other species is shown in parentheses.

**Figure 2 plants-10-01839-f002:**
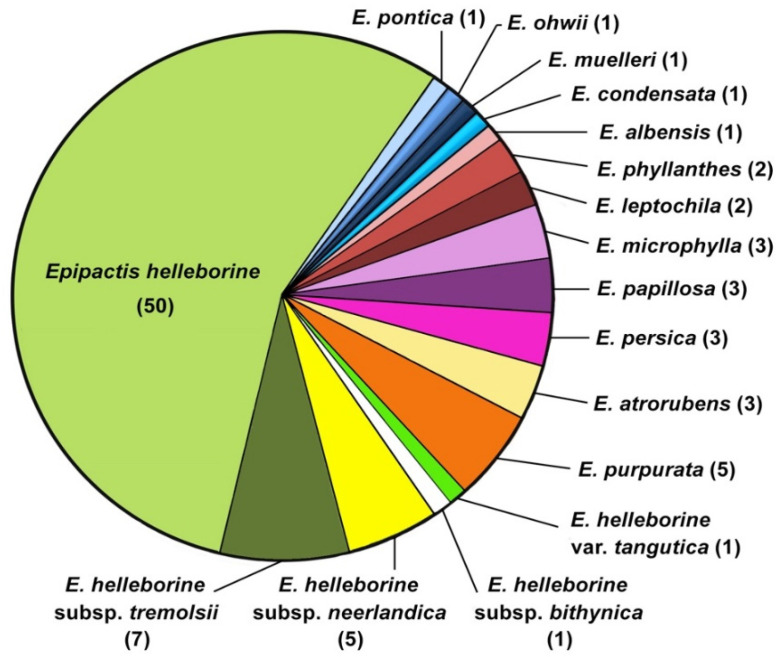
The current status of names of infraspecific taxa published in *Epipactis helleborine*, based on [73,74,75,76]. The number of names synonymised with *E. helleborine* s.l. and to other species is shown in parentheses.

**Table 1 plants-10-01839-t001:** Nomenclatural and taxonomic changes within the *Epipactis helleborine* group.

The Original Taxon Name	Inclusion in the *Epipactis* *helleborine* Group According to Different Authors						Recent Genetic Findings According to Hollingsworth et al. [4], Tranchida-Lombardo et al. [42], Sramkó et al. [24] and Bateman [11]	Currently Accepted Taxon Name According to Taxonomic Databases Available Online (31 July 2021)			
	Tyteca & Dufrêne [47]	Delforge [30]	Delforge [17]	Brullo et al. [23]	Delforge [31]	Others articles [3,5,8,10,70,71]		POWO [73]	WCSP [74]	WCVP [75]	WFO [76]
*Epipactis bugacensis* Robatsch		*×*					cannot be separated at species level from *E**pipactis dunensis* (T. Stephenson & T.A.Stephenson) Godfery	*Epipactis bugacensis* Robatsch	*Epipactis bugacensis* Robatsch	*Epipactis bugacensis* Robatsch	*Epipactis bugacensis* Robatsch
*Epipactis calabrica* U.Grabner, S.Hertel & Presser					*×*		not included in analysis	*Epipactis helleborine* subsp. *helleborine*	*Epipactis helleborine* subsp. *helleborine*	*Epipactis helleborine* subsp. *helleborine*	not included in database
*Epipactis condensata* Boiss. ex D.P.Young		*×*				*×*	not included in analysis	*Epipactis condensata* Boiss. ex D.P.Young	*Epipactis condensata* Boiss. ex D.P.Young	*Epipactis condensata* Boiss. ex D.P.Young	*Epipactis condensata* Boiss. ex D.P.Young
*Epipactis condensata* var. *kuenkeleana* (Akhalk., H. Baumann, R. Lorenz & Mosul.) Popovich						*×*	not included in analysis	not included in database	not included in database	not included in database	not included in database
*Epipactis cupaniana* C.Brullo, D’Emerico & Pulv.				*×*			not included in analysis	*Epipactis cupaniana* C.Brullo, D’Emerico & Pulv.	*Epipactis cupaniana* C.Brullo, D’Emerico & Pulv.	*Epipactis cupaniana* C.Brullo, D’Emerico & Pulv.	not included in database
*Epipactis danubialis* Robatsch & Rydlo		*×*					not included in analysis	*Epipactis atrorubens* (Hoffm.) Besser	*Epipactis atrorubens* (Hoffm.) Besser	*Epipactis atrorubens* (Hoffm.) Besser	*Epipactis atrorubens* (Hoffm.) Besser
*Epipactis densifoli**a* W.Hahn, Passin & R.Wegener			*×*	*×*	*×*		not included in analysis	*Epipactis helleborine* subsp. *tremolsii* (Pau) E.Klein	*Epipactis helleborine* subsp. *tremolsii* (Pau) E.Klein	*Epipactis helleborine* subsp. *tremolsii* (Pau) E.Klein	*Epipactis* *helleborine* subsp. *densifolia* (W.Hahn, Passin & R.Wegener) Kreutz
*Epipactis distans* Arv.-Touv.	*×*	*×*					recognised as a well-founded subspecies	*Epipactis helleborine* subsp. *helleborine*	*Epipactis helleborine* subsp. *helleborine*	*Epipactis helleborine* subsp. *helleborine*	*Epipactis* *helleborine* subsp. *orbicularis* (K.Richt.) E.Klein
*Epipactis dunensis* (T.Stephenson & T.A.Stephenson) Godfery		*×*	*×*	*×*	*×*		recognised as a genuine species	*Epipactis dunensis* (T.Stephenson & T.A.Stephenson) Godfery	*Epipactis dunensis* (T.Stephenson & T.A.Stephenson) Godfery	*Epipactis dunensis* (T.Stephenson & T.A.Stephenson) Godfery	*Epipactis dunensis* (T.Stephenson & T.A.Stephenson) Godfery
*Epipactis dunensis* var. *tynensis* (Kreutz) P.Delforge					*×*		cannot be recognised as well-founded variety	*Epipactis dunensis* (T.Stephenson & T.A.Stephenson) Godfery	*Epipactis dunensis* (T.Stephenson & T.A.Stephenson) Godfery	*Epipactis dunensis* (T.Stephenson & T.A.Stephenson) Godfery	not included in database
*Epipactis etrusca* Presser & S.Hertel					*×*		not included in analysis	*Epipactis helleborine* subsp. *helleborine*	*Epipactis helleborine* subsp. *helleborine*	*Epipactis helleborine* subsp. *helleborine*	not included in database
*Epipactis greuteri* H.Baumann & Künkele		*×*					recognised as a genuine species	*Epipactis greuteri* H.Baumann & Künkele	*Epipactis greuteri* H.Baumann & Künkele	*Epipactis greuteri* H.Baumann & Künkele	*Epipactis greuteri* H.Baumann & Künkele
*Epipactis halacsyi* Robatsch		*×*					not included in analysis	*Epipactis purpurata* subsp. *halacsyi* (Robatsch) Kreutz	*Epipactis purpurata* subsp. *halacsyi* (Robatsch) Kreutz	*Epipactis purpurata* subsp. *halacsyi* (Robatsch) Kreutz	*Epipactis purpurata* Sm.
*Epipactis helleborine* (L.) Crantz	*×*	*×*	*×*	*×*	*×*		recognised as a genuine species	*Epipactis helleborine* (L.) Crantz	*Epipactis helleborine* (L.) Crantz	*Epipactis helleborine* (L.) Crantz	*Epipactis helleborine* (L.) Crantz
*E**pipactis helleborine* subsp. *bithynica* (Robatsch) Kreutz						*×*	not included in analysis	*E**pipactis helleborine* subsp. *bithynica* (Robatsch) Kreutz	*E**pipactis helleborine* subsp. *bithynica* (Robatsch) Kreutz	*E**pipactis helleborine* subsp. *bithynica* (Robatsch) Kreutz	*E**pipactis helleborine* subsp. *bithynica* (Robatsch) Kreutz
*Epipactis helleborine* var. *castanearum* Gévaudan, Nicole & Anglade					*×*		not included in analysis	*Epipactis helleborine* subsp. *tremolsii* (Pau) E.Klein	*Epipactis helleborine* subsp. *tremolsii* (Pau) E.Klein	*Epipactis helleborine* subsp. *tremolsii* (Pau) E.Klein	*Epipactis helleborine* (L.) Crantz
*Epipactis helleborine* var. *orbicularis* (C.Richt) Soó			*×*		*×*		not included in analysis	not included in database	*Epipactis helleborine* subsp. *helleborine*	not included in database	*Epipactis helleborine* subsp. *orbicularis* (K.Richt) E.Klein
*Epipactis helleborine* subsp. *papillosa* (Franch. & Sav.) Fateryga						*×*	not included in analysis	*Epipactis papillosa* Franch. & Sav.	*Epipactis papillosa* Franch. & Sav.	*Epipacti papillosa* Franch. & Sav.	not included in database
*E**pipactis helleborine* var. *tangutica* (Schltr.) S.C.Chen & G.H.Zhu						*×*	not included in analysis	*E**pipactis helleborine* var. *tangutica* (Schltr.) S.C.Chen & G.H.Zhu	*E**pipactis helleborine* var. *tangutica* (Schltr.) S.C.Chen & G.H.Zhu	*E**pipactis helleborine* var. *tangutica* (Schltr.) S.C.Chen & G.H.Zhu	*E**pipactis helleborine* var. *tangutica* (Schltr.) S.C.Chen & G.H.Zhu
*Epipactis helleborine* var. *youngiana* (A.J.Richards & A.F.Porter) Kreutz			*×*		*×*		synonymised with *Epipactis helleborine* subsp. *neerlandica* (Verm.) Buttler	*Epipactis helleborine* subsp. *neerlandica* (Verm.) Buttler	*Epipactis helleborine* subsp. *neerlandica* (Verm.) Buttler	*Epipactis helleborine* subsp. *neerlandica* (Verm.) Buttler	*Epipactis helleborine* (L.) Crantz
*Epipactis heraclea* P.Delforge & Kreutz			×	×	×		not included in analysis	*Epipactis helleborine* subsp. *tremolsii* (Pau) E.Klein	*Epipactis helleborine* subsp. *tremolsii* (Pau) E.Klein	*Epipactis helleborine* subsp. *tremolsii* (Pau) E.Klein	*Epipactis heraclea* P.Delforge & Kreutz
*Epipactis latina* (W.Rossi & E.Klein) B.Baumann & H.Baumann		*×*					not included in analysis	*Epipactis helleborine* subsp. *tremolsii* (Pau) E.Klein	*Epipactis helleborine* subsp. *tremolsii* (Pau) E.Klein	*Epipactis helleborine* subsp. *tremolsii* (Pau) E.Klein	*Epipactis helleborine* subsp. *latina* W.Rossi & E.Klein
*Epipactis leptochila* (Godfery) Godfery		*×*					recognised as a genuine species	*Epipactis leptochila* (Godfery) Godfery	*Epipactis leptochila* (Godfery) Godfery	*Epipactis leptochila* (Godfery) Godfery	*Epipactis leptochila* (Godfery) Godfery
*Epipactis leutei* Robatsch		×					not included in analysis	*Epipactis helleborine* subsp. *helleborine*	*Epipactis helleborine* subsp. *helleborine*	*Epipactis helleborine* subsp. *helleborine*	*Epipactis helleborine* subsp. *leutei* (Robatsch) Kreutz
*Epipactis levantina* (Kreutz, Óvári & Shifman) P.Delforge					*×*		not included in analysis	*Epipactis helleborine* subsp. *tremolsii* (Pau) E.Klein	*Epipactis helleborine* subsp. *tremolsii* (Pau) E.Klein	*Epipactis helleborine* subsp. *tremolsii* (Pau) E.Klein	not included in database
*Epipactis lusitanica* D.Tyteca	*×*	*×*					cannot be separated at species level from *E**pipactis helleborine* s.str.; recognised as arguably synonymous with *E**pipactis helleborine* s.str.	*Epipactis helleborine* subsp. *tremolsii* (Pau) E.Klein	*Epipactis helleborine* subsp. *tremolsii* (Pau) E.Klein	*Epipactis helleborine* subsp. *tremolsii* (Pau) E.Klein	*Epipactis lusitanica* D.Tyteca
*Epipactis meridionalis* H.Baumann & R.Lorenz		*×*	*×*	*×*	*×*		cannot be separated at species level from *E**pipactis helleborine* s.str.;	*Epipactis helleborine* subsp. *helleborine*	*Epipactis helleborine* subsp. *helleborine*	*Epipactis helleborine* subsp. *helleborine*	*Epipactis meridionalis* H.Baumann & R.Lorenz
*Epipactis molochina* P.Delforge			*×*	*×*	*×*		not included in analysis	*Epipactis helleborine* subsp. *helleborine*	*Epipactis helleborine* subsp. *helleborine*	*Epipactis helleborine* subsp. *helleborine*	*Epipactis* *helleborine* subsp. *molochina* (P.Delforge) Kreutz
*Epipactis muelleri* Godfery		*×*					recognised as a genuine species	*Epipactis muelleri* Godfery	*Epipactis muelleri* Godfery	*Epipacti muelleri* Godfery	*Epipactis muelleri* Godfery
*Epipactis naousaensis* Robatsch		*×*					cannot be separated at species level from *E**pipactis helleborine* s.str.	*Epipactis greuteri* H.Baumann & Künkele	*Epipactis greuteri* H.Baumann & Künkele	*Epipactis greuteri* H.Baumann & Künkele	*Epipactis* *leptochila* subsp. *naousaensis* (Robatsch) Kreutz
*Epipactis neerlandica* (Verm.) Devillers-Tersch. & Devillers	*×*	*×*					recognised as a well-founded subspecies	*Epipactis helleborine* subsp. *neerlandica* (Verm.) Buttler	*Epipactis helleborine* subsp. *neerlandica* (Verm.) Buttler	*Epipactis helleborine* subsp. *neerlandica* (Verm.) Buttler	*Epipactis helleborine* subsp. *neerlandica* (Verm.) Buttler
*Epipactis nordeniorum* Robatsch		*×*	*×*	*×*	*×*		cannot be separated at species level from *E**pipactis albensis* Nováková & Rydlo	*Epipactis helleborine * subsp. *helleborine*	*Epipactis helleborine* subsp. *helleborine*	*Epipactis helleborine* subsp. *helleborine*	*Epipactis nordeniorum* Robatsch
*Epipactis olympica* Robatsch		*×*					not included in analysis	*Epipactis greuteri* H.Baumann & Künkele	*Epipactis greuteri* H.Baumann & Künkele	*Epipactis greuteri* H.Baumann & Künkele	*Epipactis* *olympica* Robatsch
*Epipactis pontica* Taubenheim		*×*	*×*	*×*			recognised as a genuine species	*Epipactis pontica* Taubenheim	*Epipactis pontica* Taubenheim	*Epipactis pontica* Taubenheim	*Epipactis pontica* Taubenheim
*Epipactis purpurata* Sm.		*×*					recognised as a genuine species	*Epipactis* *purpurata* Sm.	*Epipactis* *purpurata* Sm.	*Epipactis* *purpurata* Sm.	*Epipactis* *purpurata* Sm.
*Epipactis renzii* Robatsch		*×*					cannot be separated at species level from *E**pipactis helleborine* subsp. *neerlandica* (Verm.) Buttler	*Epipactis helleborine* subsp. *neerlandica* (Verm.) Buttler	*Epipactis helleborine* subsp. *neerlandica* (Verm.) Buttler	*Epipactis helleborine* subsp. *neerlandica* (Verm.) Buttler	*Epipactis helleborine* subsp. *neerlandica* (Verm.) Buttler
*Epipactis rhodanensis* Gévaudan & Robatsch			*×*	*×*	*×*		cannot be separated at species level from *E**pipactis dunensis* (T.Stephenson & T.A.Stephenson) Godfery	*Epipactis bugacensis* Robatsch	*Epipactis bugacensis* Robatsch	*Epipactis bugacensis* Robatsch	*Epipactis bugacensis* subsp. *rhodanensis* (Gévaudan & Robatsch) Wucherpf.
*Epipactis schubertiorum* Bartolo, Pulv. & Robatsch			*×*	*×*	*×*		cannot be separated at species level from *E**pipactis helleborine* s.str.	*Epipactis helleborine* subsp. *helleborine*	*Epipactis helleborine* subsp. *helleborine*	*Epipactis helleborine* subsp. *helleborine*	*Epipactis* *helleborine* subsp. *schubertiorum* (Bartolo, Pulv. & Robatsch) Kreutz
*Epipactis tremolsii* Pau	*×*	*×*					cannot be separated at species level from *E**pipactis helleborine* s.str.; synonymised with *Epipactis helleborine* subsp. *tremolsii* (Pau) E.Klein	*Epipactis helleborine* subsp. *tremolsii* (Pau) E.Klein	*Epipactis helleborine* subsp. *tremolsii* (Pau) E.Klein	*Epipactis helleborine* subsp. *tremolsii* (Pau) E.Klein	*Epipactis helleborine* subsp. *tremolsii* (Pau) E.Klein
*Epipactis voethii* Robatsch			×	×	×		recognised as a well-founded subspecies	*Epipactis helleborine* subsp. *helleborine*	*Epipactis helleborine* subsp. *helleborine*	*Epipactis helleborine* subsp. *helleborine*	*Epipactis helleborine* (L.) Crantz
*Epipactis youngiana* A.J.Richards & A.F.Porter		*×*					cannot be separated at species level from *E**pipactis helleborine* s.str.; synonymised with *Epipactis helleborine* subsp. *neerlandica* (Verm.) Buttler	*Epipactis helleborine* subsp. *neerlandica* (Verm.) Buttler	*Epipactis helleborine* subsp. *neerlandica* (Verm.) Buttler	*Epipactis helleborine* subsp. *neerlandica* (Verm.) Buttler	*Epipactis helleborine* (L.) Crantz

**Table 2 plants-10-01839-t002:** An overview of names of infraspecific taxa published in *Epipactis helleborine*.

The Name and Its Infraspecific Rank in *Epipactis helleborine*	The Original Name and the Year of Its Publication	Currently Accepted Name (31 July 2021) *
*E. helleborine* subsp. *aspromontana* (Bartolo, Pulv. & Robatsch) H.Baumann & R.Lorenz	*E. aspromontana* Bartolo, Pulv. & Robatsch (1996)	*E. helleborine* subsp. *helleborine*
*E. helleborine* subsp. *bithynica* (Robatsch) Kreutz	*E. bithynica* Robatsch (1991)	*E. helleborine* subsp. *bithynica* (Robatsch) Kreutz
*E. helleborine* subsp. *condensata* (Boiss. ex D.P.Young) H.Sund.	*E. condensata* Boiss. ex D.P.Young (1970).	*E. condensata* Boiss. ex D.P.Young
*E. helleborine* subsp. *confusa* (D.P.Young) H.Sund.	*E. confusa* D.P.Young (1953)	*E. phyllanthes* G.E.Sm.
*E. helleborine* subsp. *degenii* (Szentp. & Mónus) Kreutz	*E. degenii* Szentp. & Mónus (1999)	*E. helleborine* subsp. *helleborine*
*E. helleborine* subsp. *densifolia* (W.Hahn, Passin & R.Wegener) Kreutz	*E.**densifolia* W.Hahn, Passin & R.Wegener (2003)	*E. helleborine* subsp. *tremolsii* (Pau) E.Klein
*E. helleborine* subsp. *distans* (Arv.-Touv.) R.Engel & P.Quentin	*E. distans* Arv.-Touv. (1873)	*E. helleborine* subsp. *helleborine*
*E. helleborine* subsp. *helleborine*	*E. helleborine* (L.) Crantz (1769)	*E. helleborine* subsp. *helleborine*
*E. helleborine* subsp. *latifolia* (L.) Syme	*Serapias helleborine* var. *latifolia* L. (1753)	*E. helleborine* subsp. *helleborine*
*E. helleborine* subsp. *latina* W.Rossi & E.Klein	*E. helleborine* subsp. *latina* W.Rossi & E.Klein (1987)	*E. helleborine* subsp. *tremolsii* (Pau) E.Klein
*E. helleborine* subsp. *leptochila* (Godfery) Soó	*E. viridiflora* var. *leptochila* Godfery (1919)	*E. leptochila* (Godfery) Godfery
*E. helleborine* subsp. *leutei* (Robatsch) Kreutz	*E. leutei* Robatsch (1989)	*E. helleborine* subsp. *helleborine*
*E. helleborine* subsp. *levantina* Kreutz	*E. helleborine* subsp. *tremolsii* (Pau) E.Klein (1979)	*E. helleborine* subsp. *tremolsii* (Pau) E.Klein
*E. helleborine* subsp. *lusitanica* (D.Tyteca) J.-M.Tison	*E. lusitanica* D.Tyteca (1988)	*E. helleborine* subsp. *tremolsii* (Pau) E.Klein
*E. helleborine* subsp. *minor* (R.Engel) R.Engel	*E. helleborine* var. *minor* R.Engel (1994)	*E. helleborine* subsp. *helleborine*
*E. helleborine* subsp. *molochina* (P.Delforge) Kreutz	*E. molochina* P.Delforge (2004)	*E. helleborine* subsp. *helleborine*
*E. helleborine* subsp. *moratoria* Riech. & Zirnsack	*E. helleborine* subsp. *moratoria* Riech. & Zirnsack (2008)	*E. hellebori**ne* subsp. *helleborine*
*E. helleborine* subsp. *muelleri* (Godfery) O.Bolòs, Masalles & Vigo	*E. muelleri* Godfery (1921)	*E. muelleri* Godfery
*E. helleborine* subsp. *neerlandica* (Verm.) Buttler	*E. helleborine* var. *neerlandica* Verm. (1949)	*E. helleborine* subsp. *neerlandica* (Verm.) Buttler
*E. helleborine* subsp. *ohwii* (Fukuy.) H.J.Su	*E. ohwii* Fukuy. (1934)	*E. ohwii* Fukuy.
*E. helleborine* subsp. *orbicularis* (K.Richt.) E.Klein	*E. orbicularis* K.Richt. (1887)	*E. helleborine* subsp. *helleborine*
*E. helleborine* subsp. *papillosa* (Franch. & Sav.) Fateryga	*E. papillosa* Franch. & Sav. (1878)	*E. papillosa* Franch. & Sav.
*E. helleborine* subsp. *persica* (Soó) H.Sund.	*Helleborine persica* Soó (1927)	*E. persica* (Soó) Hausskn. ex Nannf.
*E. helleborine* subsp. *phyllanthes* (G.E.Sm.) H.Sund.	*E. phyllanthes* G.E.Sm. (1852)	*E. phyllanthes* G.E.Sm.
*E. helleborine* subsp. *pontica* (Taubenheim) H.Sund.	*E. pontica* Taubenheim (1975)	*E. pontica* Taubenheim
*E. helleborine* subsp. *renzii* (Robatsch) Løjtnant	*E. renzii* Robatsch (1988)	*E. helleborine* subsp. *neerlandica* (Verm.) Buttler
*E. helleborine* subsp. *schubertiorum* (Bartolo, Pulv. & Robatsch) Kreutz	*E. schubertiorum* Bartolo, Pulv. & Robatsch (1997)	*E. helleborine* subsp. *helleborine*
*E. helleborine* subsp. *transcaucasica* A.P.Khokhr.	*E. helleborine* subsp. *transcaucasica* A.P.Khokhr. (1991)	*E. persica* (Soó) Hausskn. ex Nannf.
*E. helleborine* subsp. *tremolsii* (Pau) E.Klein	*E. tremolsii* Pau (1914)	*E. helleborine* subsp. *tremolsii* (Pau) E.Klein
*E. helleborine* subsp. *troodi* (H.Lindb.) H.Sund.	*E. troodi* H.Lindb. (1942)	*E. persica* (Soó) Hausskn. ex Nannf.
*E. helleborine* subsp. *turcica* (Kreutz) Véla & Viglione	*E. turcica* Kreutz (1997)	*E. helleborine* subsp. *tremolsii* (Pau) E.Klein
*E. helleborine* subsp. *varians* (Crantz) H.Sund.	*E. helleborine* var. *varians* Crantz (1769)	*E. purpurata* Sm.
*E. helleborine* subsp. *viridans* (Crantz) O.Schwarz	*E. helleborine* var. *viridans* Crantz (1769)	*E. atrorubens* (Hoffm.) Besser
*E. helleborine* subsp. *viridiflora* (Hoffm.) O.Schwarz	*Serapias latifolia viridiflora* Hoffm. (1804)	*E. purpurata* Sm.
*E. helleborine* subsp. *viridis* Soó	*E. helleborine* subsp. *viridis* Soó (1969)	*E. helleborine* subsp. *helleborine*
*E. helleborine* subsp. *zirnsackiana* (Riech.) Riech.	*E. zirnsackiana* Riech. (2010)	*E. helleborine* subsp. *helleborine*
*E. helleborine* var. *canescens* (Irmisch) Rchb.f.	*E. microphylla* var. *canescens* Irmisch (1846)	*E. microphylla* (Ehrh.) Sw.
*E. helleborine* var. *castanearum* Gévaudan, Nicole & Anglade	*E. helleborine* var. *castanearum* Gévaudan, Nicole & Anglade (2011)	*E. helleborine* subsp. *tremolsii* (Pau) E.Klein
*E. helleborine* var. *chlorantha* Verm.	*E. helleborine* var. *chlorantha* Verm. (1949)	*E. helleborine* subsp. *helleborine*
*E. helleborine* var. *densiflora* Verm.	*E. helleborine* var. *densiflora* Verm. (1949)	*E. helleborine* subsp. *helleborine*
*E. helleborine* var. *diversifolia* Verm.	*E. helleborine* var. *diversifolia* Verm. (1949)	*E. helleborine* subsp. *helleborine*
*E. helleborine* var. *herbacea* (Lindl.) S.N.Mitra	*E. herbacea* Lindl. (1839)	*E. helleborine* subsp. *helleborine*
*E. helleborine* var. *interrupta* Beck	*E. helleborine* var. *interrupta* Beck (1890)	*E. helleborine* subsp. *helleborine*
*E. helleborine* var. *intrusa* (Lindl.) S.N.Mitra	*E. intrusa* Lindl. (1857)	*E. helleborine* subsp. *helleborine*
*E. helleborine* var. *lancifolia* (Zapal.) Bordz.	*E. viridans* var. *lancifolia* Zapal. (1906)	*E. helleborine* subsp. *helleborine*
*E. helleborine* var. *latifolia* (L.) A.Blytt	*Serapias helleborine* var. *latifolia* L. (1753)	*E. helleborine* subsp. *helleborine*
*E. helleborine* var. *laxiflora* Verm.	*E. helleborine* var. *laxiflora* Verm. (1949)	*E. helleborine* subsp. *helleborine*
*E*. *helleborine* var. *microphylla* (Ehrh.) Rchb.f.	*Serapias microphylla* Ehrh. (1789)	*E. microphylla* (Ehrh.) Sw.
*E. helleborine* var. *minor* R.Engel	*E. helleborine* var. *minor* R.Engel (1994)	*E. helleborine* subsp. *helleborine*
*E. helleborine* var. *monotropoides* (Mousley) L.Lewis	*Amesia latifolia* f. *monotropoides* Mousley (1927)	*E. helleborine* subsp. *helleborine*
*E. helleborine* var. *moratoria* (Riech. & Zirnsack) P.Delforge	*E. helleborine* subsp. *moratoria* Riech. & Zirnsack (2008)	*E. helleborine* subsp. *helleborine*
*E. helleborine* var. *neerlandica* Verm.	*E. helleborine* var. *neerlandica* Verm. (1949)	*E. helleborine* subsp. *neerlandica* (Verm.) Buttler
*E. helleborine* var. *nuda* (Irmisch) Rchb.f.	*E. microphylla* var. *nuda* Irmisch (1846)	*E. microphylla* (Ehrh.) Sw.
*E. helleborine* var. *orbicularis* (C. Richt) Soó	*E. orbicularis* C.Richt. (1887)	*E. helleborine* subsp. *helleborine*
*E. helleborine* var. *orbicularis* (K.Richt.) Verm.	*E. orbicularis* K.Richt. (1887)	*E. helleborine* subsp. *helleborine*
*E. helleborine* var. *pallens* Gaudin	*E. helleborine* var. *pallens* Gaudin (1829)	*E. helleborine* subsp. *helleborine*
*E. helleborine* var. *papillosa* (Franch. & Sav.) T.Hashim.	*E. papillosa* Franch. & Sav. (1878)	*E. papillosa* Franch. & Sav.
*E. helleborine* var. *phoenicea* Verm.	*E. helleborine* var. *phoenicea* Verm. (1949)	*E. helleborine* subsp. *helleborine*
*E. helleborine* var. *rectilinguis* Murb.	*E. helleborine* var. *rectilinguis* Murb. (1891)	*E. leptochila* (Godfery) Godfery
*E. helleborine* var. *renzii* (Robatsch) J.Claess. Kleynen & Wielinga	*E. renzii* Robatsch (1988)	*E. helleborine* subsp. *neerlandica* (Verm.) Buttler
*E. helleborine* var. *rubiginosa* Crantz	*E. helleborine* var. *rubiginosa* Crantz (1769)	*E. atrorubens* (Hoffm.) Besser
*E. helleborine* var. *sayekiana* (Makino) T.Hashim.	*E. sayekiana* Makino (1918)	*E. papillosa* Franch. & Sav.
*E. helleborine* var. *tangutica* (Schltr.) S.C.Chen & G.H.Zhu	*E. tangutica* Schltr. (1919)	*E. helleborine* var. *tangutica* (Schltr.) S.C.Chen & G.H.Zhu
*E. helleborine* var. *thomsonii* (Hook. f.) Aswal	*E. latifolia* var. *thomsonii* Hook.f. (1890)	*E. helleborine* subsp. *helleborine*
*E. helleborine* var. *thomsonii* (Hook. f.) Karthik.	*E. latifolia* var. *thomsonii* Hook.f. (1890)	*E. helleborine* subsp. *helleborine*
*E. helleborine* var. *thomsonii* (Hook. f.) R.R. Stewart	*E. latifolia* var. *thomsonii* Hook.f. (1890)	*E. helleborine* subsp. *helleborine*
*E. helleborine* var. *thomsonii* (Hook.f.) S.N.Mitra	*E. latifolia* var. *thomsonii* Hook.f. (1890)	*E. helleborine* subsp. *helleborine*
*E. helleborine* var. *varians* Crantz	*E. helleborine* var. *varians* Crantz (1769)	*E. purpurata* Sm.
*E. helleborine* var. *violacea* (Dur.-Doq.) Rchb.f.	*E. latifolia* var. *violacea* Dur.-Duq. (1846)	*E. purpurata* Sm.
*E. helleborine* var. *viridans* Crantz	*E. helleborine* var. *viridans* Crantz (1769)	*E. atrorubens* (Hoffm.) Besser
*E. helleborine* var. *viridens* A.Gray	*E. helleborine* var. *viridens* A.Gray (1879)	*E. helleborine* subsp. *helleborine*
*E. helleborine* var. *viridiflora* (Hoffm.) Bordz.	*Serapias latifolia viridiflora* Hoffm. (1804)	*E. purpurata* Sm.
*E. helleborine* var. *youngiana* (A.J. Richards & A.F.Porter) Kreutz	*E. youngiana* A.J.Richards & A.F.Porter (1982)	*E. helleborine* subsp. *neerlandica* (Verm.) Buttler
*E. helleborine* f. *alba* (Webster) B.Boivin	*E. latifolia* f. *alba* Webster (1898)	*E. helleborine* subsp. *helleborine*
*E. helleborine* f. *albifolia* M.R.Lowe	*E. helleborine* f. *albifolia* M.R.Lowe (1990)	*E. helleborine* subsp. *helleborine*
*E. helleborine* f. *brevibracteata* (Zapal.) Bordz.	*E. viridans* var. *brevibracteata* Zapal. (1906)	*E. helleborine* subsp. *helleborine*
*E. helleborine* f. *dentata* (Zapal.) Soó	*E. viridans* var. *dentata* Zapal. (1906)	*E. helleborine* subsp. *helleborine*
*E. helleborine* f. *dilatata* (Asch. & Graebn.) Soó	*E. latifolia* var. *dilatata* Asch. & Graebn. (1907)	*E. helleborine* subsp. *helleborine*
*E. helleborine* f. *gracilis* (Dageforde ex Hegi) Pauca & Morariu	*E. latifolia* f. *gracilis* Dageforde ex Hegi (1909)	*E. albensis* Nováková & Rydlo
*E. helleborine* f. *helleborine*	*E. helleborine* (L.) Crantz (1769)	*E. helleborine* subsp. *helleborine*
*E. helleborine* f. *luteola* P.M.Br.	*E. helleborine* f. *luteola* P.M.Br. (1996)	*E. helleborine* subsp. *helleborine*
*E. helleborine* f. *macrophylla* Snarskis	*E. helleborine* f. *macrophylla* Snarskis (1963)	*E. helleborine* subsp. *helleborine*
*E. helleborine* f. *minor* (R.Engel) P.Delforge	*E. helleborine* var. *minor* R.Engel (1994)	*E. helleborine* subsp. *helleborine*
*E. helleborine* f. *monotropoides* (Mousley) Scoggan	*Amesia latifolia* f. *monotropoides* Mousley (1927)	*E. helleborine* subsp. *helleborine*
*E. helleborine* f. *montana* (Zapal.) Bordz.	*E. viridans* var. *montana* Zapal. (1906)	*E. helleborine* subsp. *helleborine*
*E. helleborine* f. *obtusa* (Zapal.) Soó	*E. viridans* var. *obtusa* Zapal. (1906)	*E. helleborine* subsp. *helleborine*
*E. helleborine* f. *parviflora* (Zapal.) Bordz.	*E. viridans* f. *parviflora* Zapal. (1906)	*E. helleborine* subsp. *helleborine*
*E. helleborine* f. *przemysliensis* (Zapal.) Verm.	*E. viridans* var. *przemysliensis* Zapal. (1906)	*E. helleborine* subsp. *helleborine*
*E. helleborine* f. *remota* (Zapal.) Bordz.	*E. viridans* f. *remota* Zapal. (1906)	*E. helleborine* subsp. *helleborine*
*E. helleborine* f. *variegata* (Webster) B.Boivin	*E. latifolia* f. *variegata* Webster (1898)	*E. helleborine* subsp. *helleborine*

***** According to [73,74,75,76].

## Data Availability

Publicly available datasets were analysed in this study.

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
