# Peer review of "The Epipactis helleborine Group (Orchidaceae): An Overview of Recent Taxonomic Changes, with an Updated List of Currently Accepted Taxa"

_plants, 2021, doi:10.3390/plants10091839_

Round 1

Reviewer 1 Report

This paper looks well and I have a few comments on it.

It is not clear what is the formal criteria to include a certain taxon to E. helleborine group. You should provide either morphological diagnosis or something like "those taxa which are recognized as its members by Delforge" or "those taxa which whenever were treated as subtaxa of E. helleborine". Several criteria are also possible (with any of them being enough).

Please check the list of species in Europe (in Abstract and Discussion). At least one of them (E. tangutica) but perhaps also E. bythinica and may be also E. condensata and E. kuenkeleana are absent from Europe (depending on the borders of Europe accepted by you).

The phrase that E. condensata susbp. kuenkeleana was described in 2014 is incorrect. It is just a new combination, not the description of. This subspecies was described in 2005 as E. viridiflora subsp. kuenkeleana (Akhalk., H.Baumann, R.Lorenz & Mosul.). To complete this story you should also add that Delforge in 2015 risen it to a full species E. kuenkeleana (Akhalk., H.Baumann, R.Lorenz & Mosul.) P. Delforge. 

Please add that E. helleborine subsp. distans is treated in the databases as a synonym of E. h. subsp. orbicularis. Name "distans" has priority at species rank while "orbicularis" has priority at subspecies rank. The combination "E. h. subsp. distans" has sense only if you consider it distinct from subsp. orbicularis (but this needs further study).

Please see the attached file for further details.

Author Response

Point-by Point Response to Comments by Review 1

It is not clear what is the formal criteria to include a certain taxon to E. helleborine group. You should provide either morphological diagnosis or something like "those taxa which are recognized as its members by Delforge" or "those taxa which whenever were treated as subtaxa of E. helleborine". Several criteria are also possible (with any of them being enough).

Response: Thank you for pointing this out (oversight on our part). We have inserted the one sentence more in the Introduction (line 83-85), where we added the decisive criterion for the selection of individual taxa to the group.

Please check the list of species in Europe (in Abstract and Discussion). At least one of them (E. tangutica) but perhaps also E. bythinica and may be also E. condensata and E. kuenkeleana are absent from Europe (depending on the borders of Europe accepted by you).

Response: Thanks. Following your suggestion, we thoroughly checked all taxa in terms of their geographical distribution. As you suggested, it turned out that more than half of them we recognize outside Europe are also found in Asia (E. helleborine subsp. helleborine, E. helleborine subsp. distans, E. helleborine subsp. voethii, E. condensata, E. condensata var. kuenkeleana) and one even in Africa (E. helleborine subsp. tremolsii). Additionally, two others are exclusive to Asia (E. helleborine subsp. bithynica, E. helleborine var. tangutica). Therefore, throughout the manuscript, we have removed all previously made suggestions that all of the taxa we have included to the group should only occur in Europe.

The phrase that E. condensata susbp. kuenkeleana was described in 2014 is incorrect. It is just a new combination, not the description of. This subspecies was described in 2005 as E. viridiflora subsp. kuenkeleana (Akhalk., H.Baumann, R.Lorenz & Mosul.). To complete this story you should also add that Delforge in 2015 risen it to a full species E. kuenkeleana (Akhalk., H.Baumann, R.Lorenz & Mosul.) P. Delforge.

Response: Thank you for pointing this out (oversight on our part). The full history of taxonomic changes in relation to the Epipactis condensata subsp. kuenkeleana has been completed (line 130-138).

Please add that E. helleborine subsp. distans is treated in the databases as a synonym of E. h. subsp. orbicularis. Name "distans" has priority at species rank while "orbicularis" has priority at subspecies rank. The combination "E. h. subsp. distans" has sense only if you consider it distinct from subsp. orbicularis (but this needs further study).

Response: We cannot fully agree with your suggestion that E. helleborine subsp. distans is treated in the available taxonomic databases as a synonym of E. helleborine subsp. orbicularis. We checked it and as it turns out in three out of the four databases (i.e., POWO, WCSP, WCVP) this taxon is known as a synonym of E. helleborine subsp. helleborine. According to only the one database (i.e., WFO), it is known as a synonym of E. helleborine subsp. orbicularis. In addition, both the name „distans” and the name „orbicularis” have priority at species rank, because were used in the original names published taxa (i.e., E. distans Arv.-Touv., E. orbicularis K.Richt). In our opinion, the combination E. helleborine subsp. distans makes sense, because it was already used by Engel & Quentin in 1996 (see Epipactis helleborine subsp. distans (Arv.-Touv.) R.Engel & P.Quentin | Plants of the World Online | Kew Science) and the taxonomic legitimacy to distinguished this taxon has been genetically confirmed by Sramkó et al. in 2019.

Reviewer 2 Report

I think it is a useful review, as the taxonomy of the genus Epipactis and especiaslly of the  Epipactis helleborine group badly needed a revision.

The structure of the paper is good and the reader is not forced to wadding through deep mud, as it often happens in review papers.

Although the language is not bad, I still feel that editing of the text by a native English speaker would help and improve reradability of the paper. The authors should also carefully check the paper for misprints, e.g. "making it, difficult", where comma is redundant.

Author Response

Point-by Point Response to Comments by Review 2

Although the language is not bad, I still feel that editing of the text by a native English speaker would help and improve reradability of the paper. The authors should also carefully check the paper for misprints, e.g. "making it, difficult", where comma is redundant.

Response: Thanks. Following your suggestions, we checked carefully our manuscript for misprints and corrected them if necessary. In addition, we have improved the text and checked the English language.

Reviewer 3 Report

The paper is well written, it makes a complete examination of the Epipacttis helleborine group and its complex nomenclatural history. 
Since it is a review, no particular novelty emerges, however the authors in their conclusions circumscribe the taxa that they consider valid without however giving reasons for their choice. 
In any case it would be opportune to highlight the valid taxa by listing them in a table 3 to facilitate the reading of the text and the comparison with the other tables. 
Finally, the authors do not take a position on the autogamous species and this should be justified.

Below I highlight some oversights and typos in the text

line 32 defied probably you mean defined
line 50 referred to as  - delete as  or change with "the"
line 54 a wide range of characters delimitate and group the species - check this sentence probably you mean a wide range of characters which delimitate and group the species
line 85 But authors - probably you mean but these authors or but the authors
line 213 morfological change in morphological

Author Response

Point-by Point Response to Comments by Review 3

Since it is a review, no particular novelty emerges, however the authors in their conclusions circumscribe the taxa that they consider valid without however giving reasons for their choice.

Response: Thank you for pointing this out (oversight on our part). We have inserted the one sentence more in the Introduction (line 83-85), where we added the decisive criterion for the selection of individual taxa to the group.

In any case it would be opportune to highlight the valid taxa by listing them in a table 3 to facilitate the reading of the text and the comparison with the other tables.

Response: Thank you very much for this remark. Unfortunately, we do not agree with the reviewer’s opinion. We tried to add a table, but finally gave it up because this table duplicated the information contained in the conclusions. In our opinion, such a table is redundant in the revised version of the manuscript.

Finally, the authors do not take a position on the autogamous species and this should be justified.

Response: In our opinion, taking a final position on the position of all autogamous species in the classification system of the genus Epipactis is possible only after thorough morphological and genetic studies. As our work is a review paper, we only included published data.

We believe that any obligatory autogamous taxa should not be included in the groups devoted to allogamous taxa such as for example E. helleborine. We therefore confirm that previous exclusion of autogamous taxa such as E. bugacensis, E. leptochila, E. muelleri, as well as E. pontica by Pierre Delforge of the E. helleborine group was the right decision. Thus, taking into account the results of recent genetic analysis (Sramkó et al. 2019), we considered it necessary to excluded the last remaining autogamous taxon, i.e., E. dunensis in this group.

line 32 defied probably you mean defined

Response: Corrected.

line 50 referred to as - delete as or change with "the"

Response: Corrected.

line 54 a wide range of characters delimitate and group the species - check this sentence probably you mean a wide range of characters which delimitate and group the species

Response: Corrected.

line 85 But authors - probably you mean but these authors or but the authors

Response: Corrected.

line 213 morfological change in morphological

Response: Corrected.

Reviewer 4 Report

Review of the article entitled: “The Epipactis helleborine Group (Orchidaceae): An Overview of Recent Taxonomic Changes, with an Updated List of Currently Accepted European Taxa”

Summary

The article presents a review of the taxonomic situation concerning the Epipactis helleborine Group (Orchidaceae): composition, and taxonomic relationships. As stated in the introduction the panorama is of a high complexity and presents some confusion. In consequence, a review article that provides some light on these aspects is welcome, but commentaries below must be taken into consideration.

General comment

Some relevant aspects of the taxonomy are briefly mentioned, and a more detailed description is required. An increased critical discussion would contribute to increased interest for the readers and is strongly suggested.

The conclusions of the article include the Proposal of a new status for Epipactis helleborine subsp. voethii. This should be mentioned in the Abstract, while a summary of the group should be included only if it is given with a high precision level, greater than it is now presented (see later).

Commentaries by sections:

Abstract:

It reads:

“Because of the exceptionally high levels of morphological variability and the ability to readily cross with other species, ninety, different taxa at various taxonomic ranks have already been described within it, but the taxonomic status of most of them is uncertain and widely disputed.”

Could it be better? (Please check):

“Because of the exceptionally high levels of morphological variability and the ability to readily cross with other species, ninety different taxa at various taxonomic ranks have already been described within it, but the taxonomic status of most of them is uncertain, widely disputed, sometimes irrelevant.”

It reads:

“Based on the reviewed literature and data collected in four taxonomic databases available online, we propose an updated list of accepted European taxa in the Epipactis helleborine group, which includes: E. helleborine (L.) Crantz subsp. helleborine, E. helleborine subsp. bithynica (Robatsch) Kreutz, E. helleborine subsp. distans (Arv.-Touv.) R.Engel & P.Quentin, E. helleborine subsp. neerlandica (Verm.) Buttler, E. helleborine var. tangutica (Schltr.) S.C.Chen &  G.H.Zhu, E. helleborine subsp. tremolsii (Pau) E.Klein, E. helleborine subsp. voethii (Robatsch) Jakubska-Busse, Żołubak & Łobas, stat. nov., E. condensata Boiss. ex D.P.Young, E. condensata var. kuenkeleana (Akhalk., H. Baumann, R. Lorenz & Mosul.) Popovichand E. cupaniana C.Brullo, D'Emerico & Pulv.”

Please make clear the final composition of the group. According to this paragraph the Epipactis helleborine group is composed by six subspecies: subsp. helleborine, subsp. bithynica, subsp. distans, subsp. neerlandica, subsp. tremolsii, subsp. voethii, one variety: var. tangutica, and one species: E. condensate.

If this is correct, then it is not clear the status of E. cupaniana at the end of this paragraph.

Could you please make clearer this important paragraph? Please state clearly the number of items in each category.

  1. Introduction

Row 32: “Defied” should be “defined”

The relative importance of morphological and molecular criteria needs to be discussed. The article quoted in ref. 11 deserves a more detailed commentary. Do the authors believe that nucleotide sequencing is the fundamental tool for taxonomy? What morphological characters should be evaluated as potential clues for taxonomy? Please expand your views on these aspects.

  1. Recent taxonomic publications devoted to the Epipactis helleborine group

This section starts introducing the work by Tyteca and Dufrêne (ref. 37). The results of this work are clearly in contrast to what is proposed by the authors. For Tyteca and Dufrêne the Epipactis helleborine group is composed by 5 species: Epipactis helleborine s. str., E. distans Arv.-Touv., E. neerlandica (Verm.) Devillers-Tersch. & Devillers, E. tremolsii Pau and E. lusitanica D.Tyteca. Please discuss and provide an explanation or hypothesis to explain this difference?

Table 1 includes some aspects that deserve a detailed explanation. For example: What are the molecular methods and/or genetic findings in references 11, 24 and 42. Also it may seem contradictory the statement in this table that Epipactis distans Arv.-Touv. Is recognized as a well-funded subspecies, while it is classified by POWO, WCSP, WCVP, and WFO as Epipactis helleborine subsp. helleborine. Please could you explain this apparent contradiction?

  1. List of names of infraspecific taxa in Epipactis helleborine and its current taxonomic status

It reads:

“The seemingly endless morphological variation observed across the entire distribution range of Epipactis helleborine is clearly reflected by the list of infraspecific taxa presented below in Table 2.”

But the large list of taxa can be due to other factors different from morphological variation, for example lack of homogeneity in the criteria (multiple groups working with different methods) or poor communication means between distant scientists. It may be interesting to discuss in detail these aspects. For example: Could you provide examples of the mentioned “endless morphological variation”? Could you discuss any other alternatives to it as the source of taxonomic discrepance?

  1. Conclusions

Row 178, change analyses to analysis

Rows 179-181. Correct:

Epipactis naousaensis and E. olympica are in fact molecularly identical to E. greuteri, which in turn is considerably differs of E. helleborine by morphological characters.

To:

Epipactis naousaensis and E. olympica are in fact molecularly identical to E. greuteri, which in turn differs considerably of E. helleborine by morphological characters.

Row 213, change: morfological to morphological

Check Ref. [39]

Author Response

Point-by Point Response to Comments by Reviewer 4

It reads:

“Because of the exceptionally high levels of morphological variability and the ability to readily cross with other species, ninety, different taxa at various taxonomic ranks have already been described within it, but the taxonomic status of most of them is uncertain and widely disputed.”

Could it be better? (Please check):

“Because of the exceptionally high levels of morphological variability and the ability to readily cross with other species, ninety different taxa at various taxonomic ranks have already been described within it, but the taxonomic status of most of them is uncertain, widely disputed, sometimes irrelevant.”

Response: Indeed, you are right. We have made this correction.

It reads:

“Based on the reviewed literature and data collected in four taxonomic databases available online, we propose an updated list of accepted European taxa in the Epipactis helleborine group, which includes: E. helleborine (L.) Crantz subsp. helleborineE. helleborine subsp. bithynica (Robatsch) Kreutz, E. helleborine subsp. distans (Arv.-Touv.) R.Engel & P.Quentin, E. helleborine subsp. neerlandica (Verm.) Buttler, E. helleborine var. tangutica (Schltr.) S.C.Chen &  G.H.Zhu, E. helleborine subsp. tremolsii (Pau) E.Klein, E. helleborine subsp. voethii (Robatsch) Jakubska-Busse, Żołubak & Łobas, stat. nov., E. condensata Boiss. ex D.P.Young, E. condensata var. kuenkeleana (Akhalk., H. Baumann, R. Lorenz & Mosul.) Popovichand E. cupaniana C.Brullo, D'Emerico & Pulv.”

Please make clear the final composition of the group. According to this paragraph the Epipactis helleborine group is composed by six subspecies: subsp. helleborine, subsp. bithynica, subsp. distans, subsp. neerlandica, subsp. tremolsii, subsp. voethii, one variety: var. tangutica, and one species: E. condensate.

If this is correct, then it is not clear the status of E. cupaniana at the end of this paragraph.

Could you please make clearer this important paragraph? Please state clearly the number of items in each category.

Response: Thanks. Following your suggestion, we made clear the final circumscription of the E. helleborine group, which included within 10 taxa. In addition, the information about the number of taxa categorized according to its taxonomic ranks we added in parenthesis (line 21). Our surnames as well as the abbreviation for „status novus” after name of taxa (i.e., E. helleborine subsp. voethii (Robatsch) Jakubska-Busse, Żołubak & Łobas, stat. nov.) clearly indicate to the readers of our manuscript that we have assigned this taxa a new taxonomic status.

Row 32: “Defied” should be “defined”

Response: Corrected.

The relative importance of morphological and molecular criteria needs to be discussed. The article quoted in ref. 11 deserves a more detailed commentary. Do the authors believe that nucleotide sequencing is the fundamental tool for taxonomy? What morphological characters should be evaluated as potential clues for taxonomy? Please expand your views on these aspects.

Response: Epipactis helleborine s.str. is a morphologically highly variable taxon. Please look at the figure below. It presents a summary of the morphological variability of E. helleborine s.str. flowers from four isolated populations in Lower Silesia, where we conduct long-term, also taking into account the observations of clonal variability. There are not separate taxa, but the morphological variability of the one taxon!

Due to the differences in species identification by various specialists in the taxonomy of orchids, we are very careful about classification only on the basis of morphometry results.

Unfortunately, in our opinion, it is necessary to use extended genetic analysis, look for new molecular markers that will allow to establish the real relationship between taxa. Bateman’s article, in our opinion, deals with a different problem and has been sufficiently discussed. We are sorry, but we do not agree with the reviewer’s opinion.

This section starts introducing the work by Tyteca and Dufrêne (ref. 37). The results of this work are clearly in contrast to what is proposed by the authors. For Tyteca and Dufrêne the Epipactis helleborine group is composed by 5 species: Epipactis helleborine s. str., E. distans Arv.-Touv., E. neerlandica (Verm.) Devillers-Tersch. & Devillers, E. tremolsii Pau and E. lusitanica D.TytecaPlease discuss and provide an explanation or hypothesis to explain this difference?

Response: Contrary to appearances, the results of this research article do not contrast to what we proposed in our circumscription as seems at first glance. According to Table 1, in the light of the recent genetic findings all the five taxa included within the E. helleborine group by Tyteca & Dufrene in 1994 are still in it, although in fact the taxonomic status of these taxa have changed since then (E. helleborine s.str.= E. helleborine subsp. helleborine; E. distans = E. helleborine subsp. distans; E. neerlandica = E. helleborine subsp. neerlandica; E. tremolsii = E. helleborine subsp. tremolsii; E. lusitanica = E. helleborine subsp. helleborine). In the case of the six remaining taxa that we included within our circumscription (i.e., E. helleborine subsp. bithynica; E. helleborine var. tangutica; E. helleborine subsp. voethii; E. condensata; E. condensata var. kuenkeleana; E. cupaniana), it should be noted that in the following years they were included within the E. helleborine group by other authors. Moreover, two of them in 1994 (E. condensata var. kuenkeleana; E. cupaniana) have not even been described yet.

Table 1 includes some aspects that deserve a detailed explanation. For example: What are the molecular methods and/or genetic findings in references 11, 24 and 42. Also it may seem contradictory the statement in this table that Epipactis distans Arv.-Touv. Is recognized as a well-funded subspecies, while it is classified by POWO, WCSP, WCVP, and WFO as Epipactis helleborine subsp. helleborine. Please could you explain this apparent contradiction?

Response: The molecular methods used in the articles to which we referred in Table 1 (i.e., 4,42,24,11) are described in detail in it. Therefore, we believe that it is unnecessary to mention them again in our manuscript. We have included references to the literature cited in Table 1, so that the readers interested in our manuscript can easy to find this articles and get interesting details. However, in the case of the genetic findings you mentioned, we presents them here in the column entitled “Recent genetic findings according to…” Indeed, you are right about the taxonomic discrepancies in relation to E. distans. As it turns out, although this taxon has been genetically recognized as a well-founded subspecies by Sramkó et al. in 2019, in fact it is still know in the available taxonomic databases as E. distans. This ensures that our manuscript is needed and in it, as you probably noticed, this taxon is included by us within the E. helleborine group at the rank of subspecies, which it already had (see Epipactis helleborine subsp. distans (Arv.-Touv.) R.Engel & P.Quentin | Plants of the World Online | Kew Science).

It reads:

“The seemingly endless morphological variation observed across the entire distribution range of Epipactis helleborine is clearly reflected by the list of infraspecific taxa presented below in Table 2.”

But the large list of taxa can be due to other factors different from morphological variation, for example lack of homogeneity in the criteria (multiple groups working with different methods) or poor communication means between distant scientists. It may be interesting to discuss in detail these aspects. For example: Could you provide examples of the mentioned “endless morphological variation”? Could you discuss any other alternatives to it as the source of taxonomic discrepance?

Response: The problem of morphological variability in the genus Epipactis is well known. Our manuscript is a review paper, in which we collected information published and based on it we made a summary of the current knowledge. Introducing the discussion suggested by the reviewer would require us to present our unpublished data, which we plan to publish soon in separate article. We have left this fragment of the text unchanged.

Row 178, change analyses to analysis

Response: Corrected.

Rows 179-181. Correct:

Epipactis naousaensis and E. olympica are in fact molecularly identical to E. greuteri, which in turn is considerably differs of Ehelleborine by morphological characters.

To:

Epipactis naousaensis and E. olympica are in fact molecularly identical to E. greuteri, which in turn differs considerably of Ehelleborine by morphological characters.

Response: Indeed. We have made this correction.

Row 213, change: morfological to morphological

Response: Corrected.

Check Ref. [39]

Response: Indeed, rightly directed our attention to check it, because as it turns out this chapter was published twice. The first time was in 2014 (see Plant Taxonomy: A Historical Perspective, Current Challenges, and Perspectives | SpringerLink) and the second (with slight changes) in 2021 (see Plant Taxonomy: A Historical Perspective, Current Challenges, and Perspectives | SpringerLink). Therefore, we cited the latter ones in our manuscript.

Round 2

Reviewer 4 Report

The questions in the first review have been answered.